Review

 

**Subject Area:**
biochemistry

protein damage, ageing, age-related diseases

**Authors for correspondence:**
Anita Krisko
e-mail: anita.krisko@medils.hr
Miroslav Radman
e-mail: miroslav.radman@medils.hr

# Protein damage, ageing and age-related diseases

Anita Krisko[1] and Miroslav Radman[1,2,3]

[1]Mediterranean Institute for Life Sciences (MedILS), 21000 Split, Croatia
[2]Naos Institute for Life Sciences, 13290 Aix-en-Provence, France
[3]Inserm U-1001, Université Paris-Descartes, Faculté de Médecine Paris-Descartes, 74014 Paris, France

Ageing is considered as a snowballing phenotype of the accumulation of damaged dysfunctional or toxic proteins and silent mutations (polymorphisms) that sensitize relevant proteins to oxidative damage as inborn predispositions to age-related diseases.

Ageing is not a disease, but it causes (or shares common cause with) age-related diseases as suggested by similar slopes of age-related increase in the incidence of diseases and death. Studies of robust and more standard species revealed that dysfunctional oxidatively damaged proteins are the root cause of radiation-induced morbidity and mortality. Oxidized proteins accumulate with age and cause reversible ageing-like phenotypes with some irreversible consequences (e.g. mutations). Here, we observe in yeast that aggregation rate of damaged proteins follows the Gompertz law of mortality and review arguments for a causal relationship between oxidative protein damage, ageing and disease. Aerobes evolved proteomes remarkably resistant to oxidative damage, but imperfectly folded proteins become sensitive to oxidation. We show that α-synuclein mutations that predispose to early-onset Parkinson's disease bestow an increased intrinsic sensitivity of α-synuclein to *in vitro* oxidation. Considering how initially silent protein polymorphism becomes phenotypic while causing age-related diseases and how protein damage leads to genome alterations inspires a vision of predictive diagnostic, prognostic, prevention and treatment of degenerative diseases.

## 1. Introduction

While powerful technologies led to a rapid growth of a descriptive molecular cell biology and to the development of new drugs, the benefit to human health remains dismal to modest. Studies of the consequences of advanced ageing and age-related diseases (ARD), rather than their causes, are not likely to lead to their mitigation. Molecular details accumulate in the absence of critical phenomenological studies such that the bottleneck to the progress of biomedical science appears more at the level of concepts than technologies. Here, we propose simple concepts about plausible common causes and mechanisms of ARD, and ageing itself, and avoid consideration of 'mechanistic studies' of late-stage diseases and ageing. However, they can be readily accommodated within the framework of the proposed concepts.

Ageing is age-related acceleration of the degradation of cellular and tissue homeostasis causing malfunction, morbidity and death predisposed by inborn 'weak links'. The incidence of nearly all human ARD and death increases exponentially with age, with similar slopes (about fifth power of time) suggesting a plausible common root cause, termed intrinsic ageing [1]. Here, we enquire about the existence of a basic chemistry of intrinsic ageing 'clock' (subjected to extrinsic influences) and explore which biological substrates are the inborn weak links predisposing to particular ARD. We conclude that oxidative protein damage is a likely common cause of ageing and ARD emerging via a common mechanism with 'snowballing' phenotypic consequences (symptoms).

royalsocietypublishing.org/journal/rsob   Open Biol. 9: 180249

The literature on biology of ageing and reactive oxygen species (ROS) is exhaustive and exhausting. While countless correlations between ROS, oxidative stress, ARD and ageing cannot be chance products, there is no sign of a conceptual 'home run'. Contradictory conclusions about biological effects of ROS range from obviously deleterious (e.g. [2]) to vitally important (e.g. [3]). But adopting ROS activity during evolution into some useful biological pathways does not make ROS harmless. The identity, extent, timing and location of oxidized molecules are of essence for the understanding of biological (phenotypic) effects of ROS.

When, in the evolutionary past, atmospheric oxygen rose 20-fold and energy-efficient oxidative phophorylation made its way into eukaryotic cells, the adaptation to noxious levels of ROS became the condition of survival. The severity of primeval biological effect of ROS can be appreciated by observing rapid death of obligate anaerobic bacteria and archaea exposed to atmospheric oxygen, unadapted to high ROS levels in their oxygen-free environments. The fact that addition of standard antioxidants to growth medium allows for survival and aerobic growth of obligate anaerobes [4] shows that ROS activity kills non-adapted cells. This cytotoxic property of ROS was adopted in apoptotic cell death (review [3]) and bacterial killing by the specialized cells of the immune system [5].

This is a hybrid paper—a select short review complemented by experimental data in support of new concepts that define why and how particular oxidative damage underlies ageing and ARD [6–8]. We posit that variable patterns of oxidative proteome damage generate the variety of progressing but largely reversible cellular ageing phenotypes. Ageing and its countless manifestations, including ARD, appear as snowballing phenotypes—functional consequences of age-related accumulation of oxidative damage to the vulnerable components of the proteome. We further propose that ARD can emerge as phenotypes of damage to particular disease-related proteins sensitized to oxidation by subtle structural alterations caused by silent mutations (polymorphisms).

Phenotypic complexity of advanced ageing and diseases clearly reflects the complexity of the healthy organism and not of the cause of ageing and disease. We argue that the root cause of ageing is simple (i.e. the accumulating oxidative proteome damage) with functional (phenotypic) consequences increasing with time in intensity and complexity. This concept, which was put forward by Stadtman and colleagues [8], is now further refined and supported by data reviewed or displayed in this paper showing that:

(i) Ageing-related phenotypes can emerge, progress or regress, solely at the level of protein damage, without the necessity for DNA alterations, although they inevitably occur as the consequence of oxidative damage to proteins dedicated to DNA maintenance [9].

(ii) The principal determinant of protein damage in aerobes is the evolved intrinsic protein resistance to oxidative damage, more so than the variation in levels of ROS (figures 2 and 3).

(iii) The resistance of native proteins to oxidative damage is fragile, since it can be lost by random errors in biosynthesis and inaccurate folding (figures 1 and 2 and [9–11]) as well as by consistent folding imperfections due to silent amino acid substitutions (figure 7) that are part of global protein polymorphism. The identification of such mutational polymorphisms would break new

ground in the area of predictive diagnostics of predispositions to ARD and inspire the design of interventions, at the level of proteins, for their delay or even reversion.

# 2. Review, concepts and experiments

## 2.1. Protein maintenance underlies maintenance of life

Maintenance of life requires renewal of proteins, and the renewal of cells maintains functional organs that make up healthy organisms. Because of this hierarchy and the fact that the lifetime of proteins is generally much shorter than the lifetime of cells and organisms, maintenance of protein activities underlies maintenance of life. Phenotypic change is due to altered protein activity that can be affected directly at protein level by physiological and non-physiological modifications, such as oxidation. Therefore, we consider here only the maintenance of cellular fitness via the maintenance of proteome fitness and its homeostasis facing chronic exposure to ROS generated by oxidative metabolism and acute oxidative stress by UV light, chemicals and ionizing radiation. Unlike after acute oxidative stress, observed patterns of spontaneous oxidative proteome damage are 'snapshots' of the momentary equilibria of incurred protein damage and clearance of damaged proteins by proteasome activity and/or autophagy and the compensatory *de novo* protein biosynthesis (figures 1 and 6).

We argue that snowballing phenotypes of ageing and diseases can be the downstream phenotypic consequences of proteome dysfunction caused by observed accumulation of damaged proteins. Particular functional deficits, rare on per-cell basis (e.g. cancer), involve acquired somatic mutations and/or alterations in DNA methylation pattern occurring as the consequences of damage to the relevant dedicated proteins [9,12].

## 2.2. Physiological versus toxic protein modifications

It is estimated that about 90% of functional protein diversity stems from numerous physiological post-translational modifications (PTM). Such physiological PTMs, like phosphorylation, acetylation, methylation, mono- and poly-ubiquitination, farnesylation, etc., result from enzymatic activities and are enzymatically reversible. Persisting non-physiological protein modifications, such as non-reparable oxidative protein carbonylation, are irreversible and mostly deleterious to protein activity, and expectedly to their interactions with partner molecules. Such non-physiological PTMs presumably interfere with physiological PTMs. For instance, the amino acid lysine in proteins is subject to the largest variety of physiological PTMs and is also among the most frequently carbonylated amino acids. Interference between two kinds of PTMs would have complex and varying phenotypic consequences. Severe cytotoxic effects of particular oligomeric structures formed by misfolded oxidized proteins will be discussed below.

## 2.3. Folding and stability determine intrinsic resistance of proteins to oxidative damage

We have studied functional, phenotypic consequences of the variability in total oxidative proteome damage in bacteria [9]. To alter exclusively the levels of oxidative damage to proteins

royalsocietypublishing.org/journal/rsob Open Biol. 9: 180249

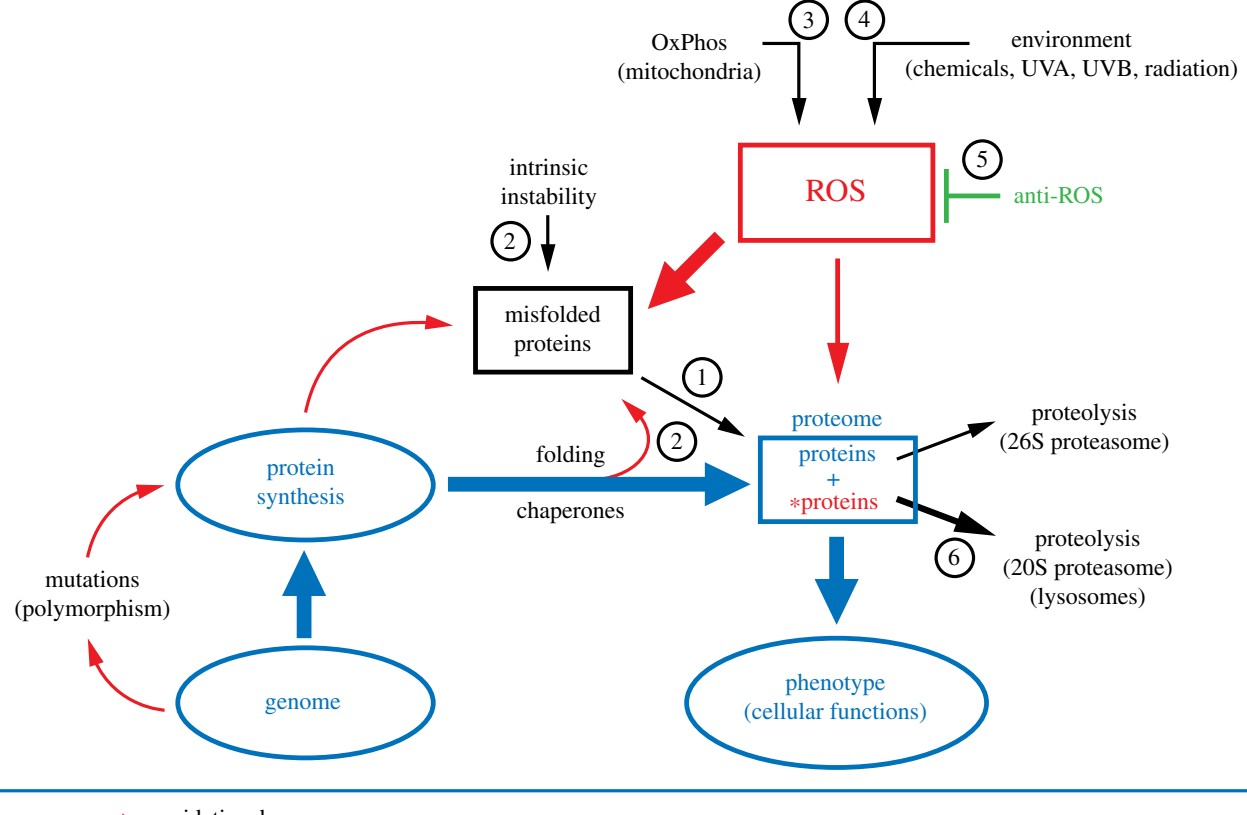

**Figure 1.** Emergence of phenotypes of oxidatively damaged proteins in relation to cellular proteostasis (from [36]). Canonical pathways of protein synthesis and folding are represented in blue. The sources and causes of oxidative protein damage are in red. Total oxidative proteome damage is a function of both ROS level and the fraction of misfolded proteins representing effective target size for protein oxidation. In addition to rare mutations resulting from malfunction of damaged proteins, cellular phenotype is determined by the extent of oxidative proteome damage and the susceptibility of individual proteins to such damage.

at constant ROS, we altered *in vivo* the susceptibility of proteins to spontaneous oxidation [10,11,13]. Such alterations were generated in *Escherichia coli* by changing the expression of three evolutionarily conserved chaperones acting on different levels of protein folding (Tig, DnaK/DnaJ and GroES/EL) and using ribosomal translational fidelity mutants *rpsL141* (high fidelity) and *rpsD14* (low fidelity) [9].

Increasing UVC irradiation of these strains revealed variations in kinetics and saturation levels of proteome oxidation (figure 2). As the survival of irradiated cell population approaches zero, the level of irreparable oxidative protein damage (protein carbonylation) reaches saturation. However, the saturation levels of protein carbonylation and cell survival after exposure to UVC differ: strains with high accuracy of protein synthesis and folding are resistant to UVC, display low constitutive proteome carbonylation and a decrease in its saturation levels relative to the wild-type [9]. The reciprocal applies as well (figure 2 and chaperone overexpression in [9]).

It appears that deviations from the native structure determine proteome 'target size' (the subpopulation of proteins sensitive to carbonylation; figure 6) for biological damage by oxidation. Since the emergence of high ROS levels (2.5–3 billion years ago), proteins were under strong selective pressure to acquire functional longevity leading to the evolution of functional structures resistant to oxidative damage. Therefore, strains from figure 2 and chaperone-down and -up mutants display expected distinct phenotypic differences for all tested biological endpoints in proportion to their protein oxidation levels [9].

## 2.4. Competitive antagonism between protein folding and oxidation and its phenotypic consequences

The described results relate to an early proposal by Kurland [14] that decreased cellular fitness, via decreased translational accuracy, may be a consequence of the impact of missense errors on protein structure and function. Novel mechanistic aspects emerged largely from Nystrom's laboratory, and later on from our laboratory, by showing that (i) misfolding predisposes proteins to oxidative damage [9–11,15] (see also figures 1–3), (ii) protein oxidation precedes aggregation and vast majority of carbonylated proteins are found in the form of aggregates [10,16], and (iii) oxidative damage to misfolded proteins causes phenotypic changes since such deleterious phenotypes can be suppressed and reversed by an antioxidant (Trolox—water-soluble vitamin E) in proportion to the decrease in protein carbonylation [9]. Even the 'visibility' of misfolded proteins by the chaperones requires their oxidation, presumably by fixing the misfolded structure, since the heat shock response to misfolding depends on oxidation of the misfolded proteins [15]. The truncated proteins cannot be folded and do not require oxidation to elicit chaperone induction [15].

Thus, it appears that oxidation (e.g. irreparable carbonylation) of misfolded proteins locks in their misfolded structures, preventing the refolding of damaged proteins by chaperones and leading to phenotypic expression of dysfunctional stably malfolded proteins (figure 3). Hence, there is a competitive antagonism between chaperone and ROS activities at the level of their common substrate: the misfolded

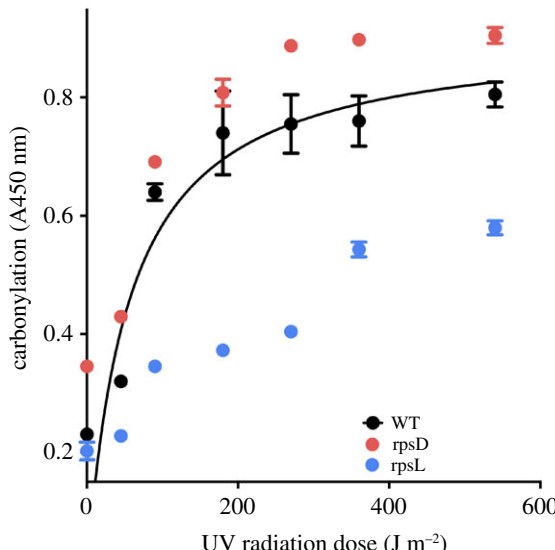

**Figure 2.** Fidelity of translation determines the target size of proteome oxidation. Protein carbonylation (measured by ELISA-based assay) was induced by increasing doses of UVC radiation up to the saturation level. Saturation level represents the proteome target size for oxidative damage and correlates with spontaneous protein oxidation of non-irradiated cells. Streptomycin (not shown) and the rpsD (*rpsD14*) mutant were used to increase translational error rates. rpsL is *rpsL141* mutant characterized by a reduced translational error rate. *Escherichia coli* was harvested from LB medium in mid-exponential growth stage, resuspended in 10 mM PBS, pH 7.4, and irradiated on ice by increasing doses of UV. Cells were collected and washed prior to protein extraction in 10 mM PBS, pH 7.4, in the presence of 1 mg ml$^{-1}$ of lysozyme and protease inhibitors (Roche). Protein concentration was determined using the Bradford assay and the total protein carbonylation was measured as described previously [9].

proteins. Apparently, native folding prevents oxidation and oxidation precludes native folding (figure 3). The observed reversibility of deleterious phenotypes caused by damaged proteome (short of mutations fixed by the mutator phenotype of proteome damage) by the antioxidant Trolox [9] is expected from a protein-based phenotypic change.

The damage–malfunction scenario might be just one part of the story. 'Malfunction' of an oxidized protein (figure 3) could sometimes mean altered function or cytotoxic function. To appreciate the conceptual significance of figure 3, the recommended papers are [9] and [15].

Protein oxidation is expected to interfere with physiological PTM and, sometimes, act itself as a physiological modification. Such hypothetical ROS-mediated 'physiological' PTM, affected by sequence polymorphism (below), could play regulatory roles upon (or by) the metabolism in cell signalling and in determination of differentiation pathways [17]. Perhaps, it is not accidental that chaperones and elongation factors are exquisitely sensitive proteomic targets for carbonylation, from bacterial to human cells [18–21]. Potential regulatory aspects of protein damage, in particular chaperone oxidation [22], are an open area for research.

The loss of global macromolecular biosynthesis, increased mutation rates and sensitivity to damage by radiation are the demonstrated phenotypes of *exclusive* oxidative damage to bacterial proteins [9]. These are also characteristic phenotypes of ageing across species, including human, whereby protein carbonylation increases quasi-exponentially with person's age [8] similarly to the increase in ARD and death rates. This raises a conceptual question: since the phenotypes of

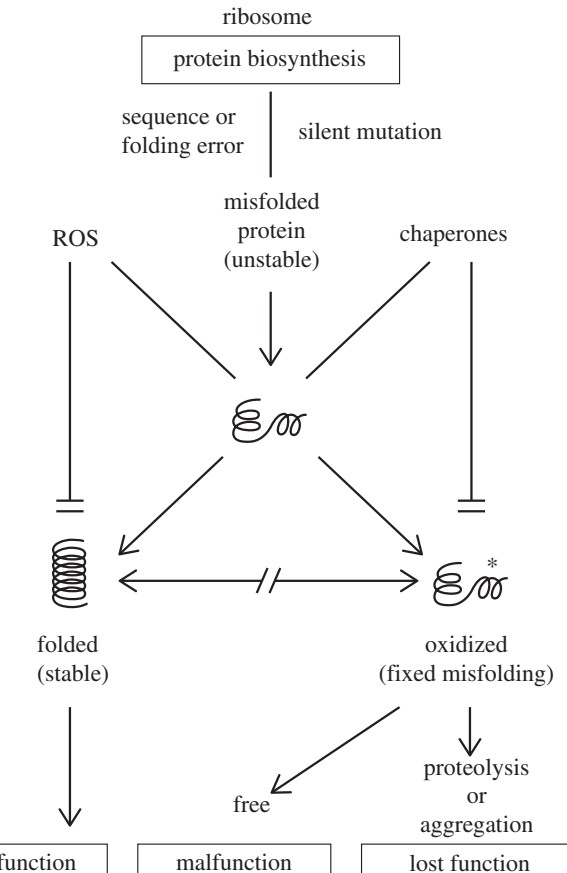

**Figure 3.** Competitive antagonism between protein folding and oxidation. Selection for functional longevity of proteins led to the evolution of oxidation-resistant structures. Imperfections in native structure arise from random biosynthetic and folding errors and from silent mutations (polymorphisms). Native folding prevents oxidation and oxidation precludes correct folding. Oxidized proteins either malfunction or lose function by aggregation or proteolysis. In particular cases, oxidation could change protein function (i.e. 'malfunction' can become a gain of function, including the 'non-self' antigenicity). In all cases, there are functional (phenotypic) consequences of protein damage via direct or cascading (snowballing) effects that are reversible by antioxidants (i.e. phenotypes of misfolding are due to oxidative damage more than to misfolding itself [9,10]).

oxidative proteome damage mimic basic ageing phenotypes, is ageing the phenotype of the proteome damage?

## 2.5. Protein misfolding and oxidation as the missing link between TOR and ROS

The fact that misfolding of erroneous proteins synthesized by low-fidelity ribosomes increases intrinsic sensitivity of proteins to oxidative damage (figure 2), and reciprocally [9], has far reaching implications for longevity and health. At the evolutionary scale, species-specific extension of longevity apparently coevolved with increased translational fidelity: there is a correlated 10-fold range of longevity and fidelity of protein biosynthesis among 17 rodent species [23]. While it is difficult to live longer by becoming another species, there is a potent physiological tuning of translational fidelity via the regulation of the speed of translation by TOR signalling: the faster the translation, the higher is its error rate. This basic postulate of the kinetic proofreading theory [24,25] holds also for transcription and replication.

At the cellular level, mTOR modulates ageing and ARD, such as diabetes type 2 and cancer, in response to energy supply (glucose), nutrients (amino acids), growth factors, oncogenes and stress. Activated mTOR pathway accelerates translation, decreases its fidelity and increases misfolding that leads to reduced protein stability [26] and expectedly (figure 3) to augmented sensitivity to oxidation (figure 6, and ongoing experiments). The finding that phenotypic effects of misfolded proteins depend on fixation of misfolded structures by oxidative damage [9,15] (figure 3) provides the conceptual 'missing link' between translational fidelity, protein misfolding and oxidation, and TOR signalling. Thus, Blagosklonny's 'ROS or TOR' [27] becomes 'ROS and TOR' whereby the link between ROS and TOR is created by augmented oxidation of misfolded proteins synthesized under activated TOR regime (figure 6). The biological effects of rapamycin, metformin and antioxidants upon ageing and several ARD can now be better understood at the molecular level.

## 2.6. Oxidative proteome damage determines spontaneous and induced mutation rates

Dedicated proteins synthesize, equilibrate and sanitize dNTP pools for DNA synthesis. They repair and replicate DNA and correct replication errors by mismatch repair. Since the efficacy and precision of numerous proteins dedicated to DNA maintenance determine the quality of DNA, it was not surprising to find that proteome damage is highly mutagenic [9]. Spontaneous mutation rates in *E. coli* correlate with roughly seventh power of proteome carbonylation. Likewise, reducing solely protein oxidative damage reduces mutation rate in *E. coli* by at least fivefold, identifying protein damage as the principal determinant of spontaneous mutation rates [9].

UVC-induced mutation frequencies correlate with constitutive and UVC-induced protein carbonylation rather than DNA damage inflicted by UVC—but do correlate with the residual (unrepaired) mutagenic DNA damage [9]. This means that oxidative damage to the DNA maintenance proteome limits its efficacy and fidelity, and thereby determines the rates of induced and spontaneous mutations [9]. Thus, unrepaired and misreplicated DNA is one of countless phenotypic consequences of proteome damage. Similar conclusions were drawn from the studies of repair of UVA and UVB damage in human skin cells [28]. Therefore, protein damage is likely to be involved—via its effects on DNA—in the initiation step of cancer and other ARD. Cancer and ARD promotion is analysed in the companion paper [29].

## 2.7. Protein damage predicts lifespan and death

There is a remarkable correlation, observed across diverse species, between the biological age (fraction of lifespan), biological fitness/performance and global protein carbonylation [8,9,30–33]. Furthermore, increased protein carbonylation is found to be associated with a variety of chronic and fatal ARD that are associated with ROS-generating chronic inflammatory conditions [34,35].

These associations inspire the question: is protein damage the cause, the consequence or a correlate (biomarker) of cellular degeneracy? After decades of serving as a biomarker of oxidative stress, protein carbonylation emerges now as a marker of the quality of cell-wide proteome folding under physiological

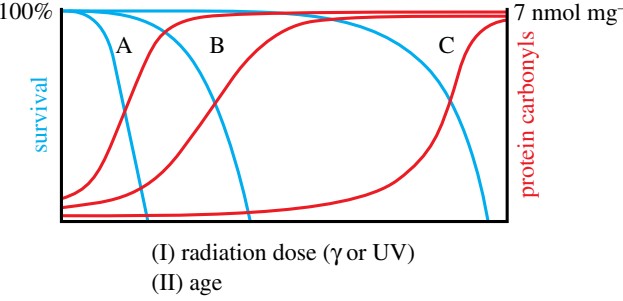

**Figure 4.** Protein carbonylation correlates with death due to radiation and living. (I) Radiation: The plots on the figure are schematized versions of published data [30,31]. Curves marked by A represent 'standard' species (*E. coli*, *C. elegans*); B represents evolved radiation-resistant *E. coli* and bdelloid rotifer *Adineta vaga* and C extremely radiation-resistant species, *D. radiodurans*. (II) Age: Protein carbonylation and survival versus age. A, B and C can be nematode, mouse and human cells. The crossings between A, B and C are schematized and are not in proportion on *x*-axis (from [36]).

conditions [9–11]. Since the misfolded proteins are particularly sensitive to oxidation, protein carbonylation can be used as a probe for 'quality of folding' at the level of individual proteins (figures 1–3) [10] *in vitro* and *in vivo*, as well as for probing the entire proteome from tissue biopsies (figure 6).

Whereas DNA damage incurred by radiation is similar in standard and extremely resistant bacterial (reviewed in [37]) and animal [31] species, the difference in radiation-induced oxidative protein damage accounts also for the differences in DNA repair and survival [9,30,31,36]. Since cell death is diagnosed hours or days after radiation of bacteria and small animals, protein damage inflicted during radiation (and measured immediately after) appears to predict death [30,36] rather than being its consequence. Studies of DNA-repair-deficient mutants displaying high mortality at low levels of proteome damage from low radiation exposures excluded the possibility that the incurred protein carbonylation could be the consequence of cell death [30].

Rather, it was shown that a common quantitative correlation exists between protein carbonylation and cell mortality across bacterial and invertebrate species—regardless of the source of radiation and species' inherent radiation resistance status (fig. 3 in [36], compiled from [30,31]; see also figure 4). This common correlation establishes oxidative protein damage as the root cause of cell death or the 'chemistry of death' [36]. The striking similarity between survival curves versus radiation, or versus lifetime, instigated the question: does the correlation between protein carbonylation and killing by exposure to radiation (fig. 1 in [30]) also stand for 'exposure to life' (i.e. radiation resistance versus chrono-resistance)? The answer is yes (figure 4) [8,36]. Cell death diagnosed by membrane permeability seems to occur as the direct consequence of protein misfolding fixed by carbonylation (figure 3) that leads to the formation of cytotoxic pore-like hydrophobic structures that insert into biological membranes ([38,39] and references therein).

## 2.8. Accumulation of damaged proteins follows the Gompertz law

The Gompertz law of mortality defines ageing as an exponential increase in the probability of death (i.e. death rate)

royalsocietypublishing.org/journal/rsob  Open Biol. **9**: 180249

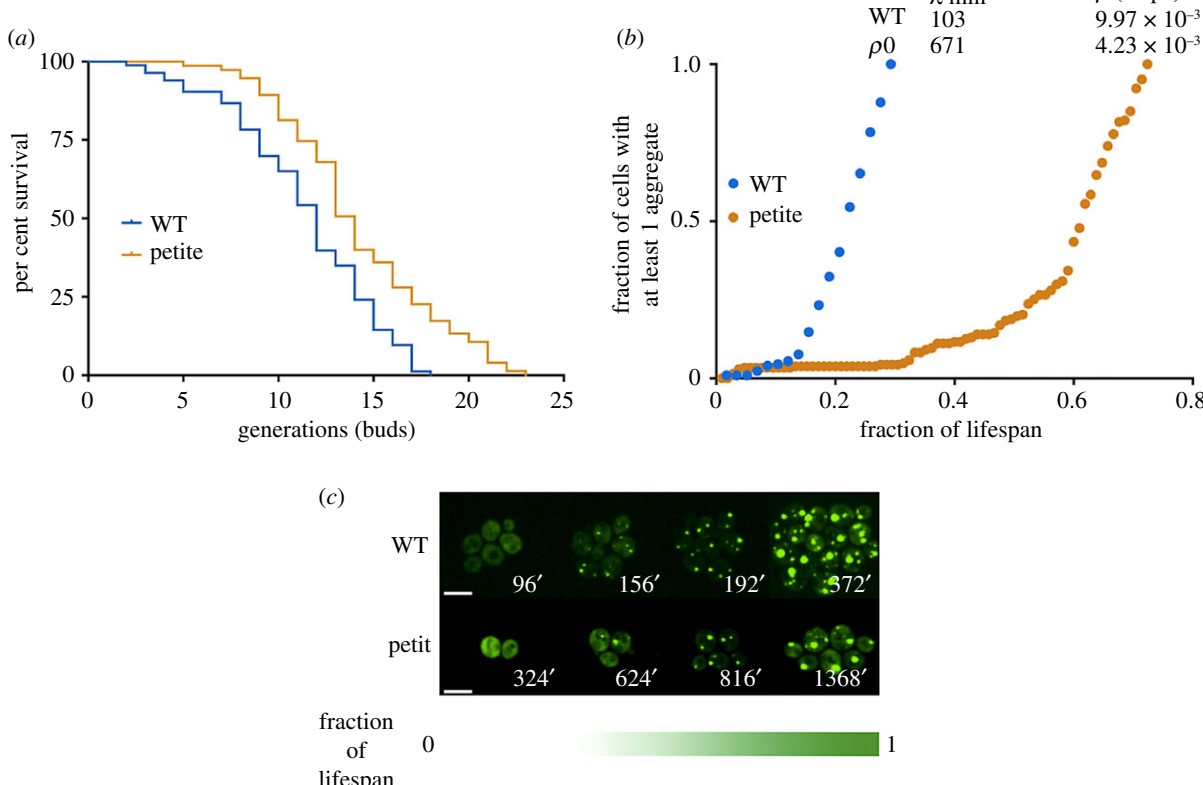

**Figure 5.** Protein aggregate appearance in budding yeast follows the Gompertz law of mortality. (a) The replicative lifespan is presented as the number of buds produced by individual mother cells. The number of cells is 103 and 122 for the wild-type and petite ($\rho0$), respectively. The data shown are pooled from two independent experiments for each strain. Significance of the results was tested with log-rank test, $p < 0.05$. Replicative lifespan was measured as described previously [43]. (b) The figure displays the increase in the fraction of cells bearing at least one aggregate. ($\lambda$) and ($\mu$) parameters from the Gompertz equation are indicated on the figure. (c) For illustration, spinning disc confocal microscopy images displaying representative images from different stages of lifespan (on the microscope slide, 30°C on YPD-rich medium). The exact time (min) at which a snapshot was taken is indicated on each picture. The white line represents 8 µm. See §5 for the description of methods related to this figure.

with age. Such increase is preceded by a lag period with no death (i.e. a constant zero mortality rate, in a protected environment), and followed by exponentially increasing late-life mortality rate until the extinction of the cohort. These are the main features of the Gompertz law of mortality, which applies from unicellular species, like budding yeast, to complex organisms like mouse and human.

Due to the lack of reliable methods to quantitate *in vivo* protein carbonylation in single cells, we applied a well-established methodology of monitoring Hsp104-GFP aggregates during lifespan of budding yeast, *Saccharomyces cerevisiae*, one of the best-described ageing-related phenotypes [40,41]. Protein aggregates consist of damaged proteins that are secluded into insoluble particles [40] shown to contain nearly all persisting carbonylated proteins in bacterial [42], yeast [40] and mouse [16] cells. Regarding protein aggregation in ageing, different studies made only qualitative observations, while quantitative analyses of protein aggregates as a reporter system for population ageing at single-cell level were missing.

Here, we confronted the replicative lifespan data for the wild-type *S. cerevisiae* and its respiration-deficient (petite) mutant with the protein aggregation propensity at single-cell level during the lifespan of both strains. We have observed that, just as the lifespan curve, the age-related dynamics of protein aggregation also follows the Gompertz law. The initial increase in mortality rate (i.e. the onset of ageing) coincides with the first observed protein aggregation events. Thus, analysis of protein aggregates at single-cell level

can be used as a marker of fitness of a cell population as well as a reporter of the biological age.

In contrast with other well-known hallmarks of ageing, like telomere shortening or genomic instability, we show that under optimal growth conditions, the appearance of protein aggregates, reflecting the accumulation of damaged proteins [16,42], follows the Gompertz function (figure 5).

## 2.9. Ageing as the phenotypic consequence of protein damage

Now, one can imagine a wealth of emerging phenotypes originating from the diversity in extent and pattern of oxidative proteome damage and formulate a hypothesis of a common basic mechanism of ageing and ARD. The basic predictions of the concept of phenotypic effects of protein damage in ageing, such as variability, progression and reversibility, have already been tested.

Variability and progression of ageing phenotypes at both population and cellular level are common knowledge. We have shown that the deleterious, yet reversible, bacterial phenotypes akin to cellular ageing (e.g. reduced biosynthetic capacity and increased mutation rates) can emerge and progress to lethality (proteomic catastrophe) solely as the consequence of accumulated protein oxidation [9].

In late 1980s, Stadtman and colleagues [8] showed that protein carbonylation accumulates in cultured human primary

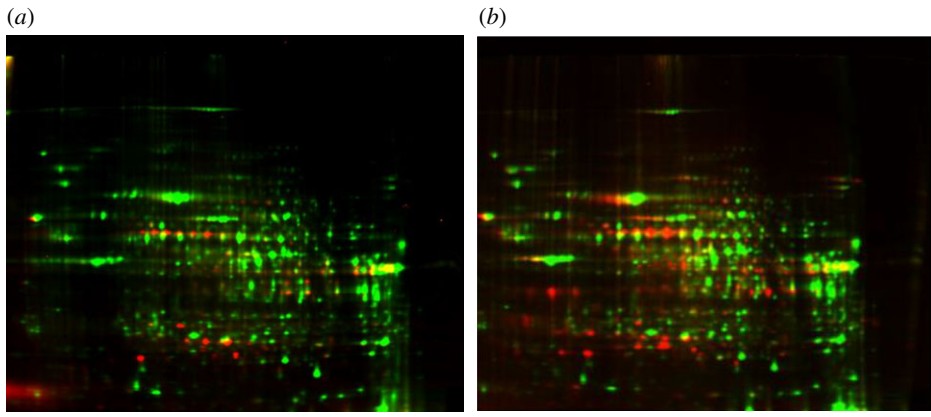

**Figure 6.** A small fraction of human liver proteome is sensitive to protein carbonylation. (a) A 2D-Oxi-DIGE 'carbonylome' of cell extracts from a human liver biopsy displays protein spots (in green) and carbonyls (in red). (b) A biopsy of a hepatocarcinoma within the same liver dispays increased cellular protein carbonylation as was found with 58 human tumours without exception. The row of seven spots in the middle of the left gel corresponds to the isoforms of the same protein with increasing negative charges due to increasing phosphorylation and carbonylation, underpinning the likely interference of different PTM. (Provided by Fernando A. Martin and Romain Ladouce, MedILS proteomic platform.)

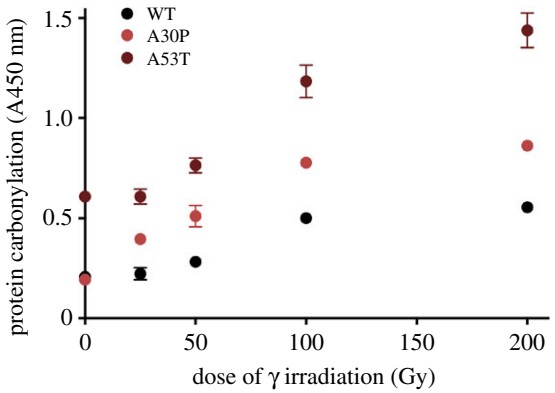

**Figure 7.** Wild-type version of α-synuclein is characterized by the highest oxidation resistance. The plot displays the differential resistance to γ radiation of human α-synuclein protein isomorphs. All three proteomorphs of α-synuclein were purchased from rPeptide. Purified proteins in PBS were exposed to increasing doses of γ radiation on ice and their carbonylation level was measured as previously described [9].

skin fibroblasts quasi-exponentially with donor's age. The increase in the fraction of carbonylated proteins (measured also directly in entire small animals and in *ex vivo* rodent and human tissues) with the fraction of lifespan of four very different species (from nematode to human) is remarkably similar [8], although the lifetimes vary from couple of weeks to hundred years. By cautiousness, Stadtman's correlations were considered to mean that protein carbonylation is a mere biomarker of oxidative stress during ageing.

Observations relating to the prediction of death by protein carbonylation levels were made with isogenic and isochronic populations of the nematode *C. elegans* [32]. A population of the same genotype and the same (young) age, maintained in the same environment, was fractionated according to the speed of animal movement (while racing unidirectionally in a weak electric field), which allowed arbitrary separation into three subpopulations (fast, intermediate and slow). Total animal protein carbonylation correlated with the speed of animal movement, whereby the fastest animals displayed the lowest protein carbonylation. The subpopulation of the fastest animals was the one with the longest remaining lifespan, whereas slower worms with intermediate and highest protein

carbonylation exhibited shorter lifespans in proportion to the measured carbonylation. Remarkably, the lifespan of fast nematodes is nearly double that of slow nematodes—at presumably constant genotype and environment. Hence, the total protein carbonylation level at young age appeared predictive of the duration of the remaining life in isogenic nematodes. Whereas protein damage in these young post-mitotic animals is unlikely to be the consequence of their performance, or of cell death, the pre-existing differences in protein carbonylation (and other oxidative damage proportional to carbonylation) appear as the biomarker or even the root cause of both endpoints. The observed variation in nematodes' protein carbonylation is probably related to the fidelity of protein biosynthesis and folding since, in the same experiment, nematodes' chaperone and carbonylation levels were correlated—as in bacteria [9,10].

This research instigates a novel paradigm in ageing research by considering ageing-related morbidity and mortality as the consequence of cellular dysfunction linked causally to the accumulation of protein damage. It becomes obvious that protein carbonylation measures biological age and predicts the remaining lifespan by reporting on reduced performance of the damaged proteome causing age-related degeneracy. Other phenotypes recognized as the hallmarks of ageing [44], like telomere attrition and genome instability, can be viewed as downstream consequences of proteome damage. For instance, telomerase is among the least abundant cellular proteins and is among the human proteins most sensitive to carbonylation [45].

## 2.10. Immortality correlates with constant levels of protein damage

The absence of ageing is defined by constant mortality rate with increasing age. If this holds for individual cells, and protein carbonylation were a faithful biomarker for predicting cell death, then immortal, or immortalized, cell lines should show constant levels of protein carbonylation. Low constitutive levels of protein carbonylation were already found in iPS cells [46]. In table 1, we show the comparison of total protein carbonylation between mouse ES cells, embryonic fibroblasts (MEFs), adult mouse skin fibroblasts and derived iPSC, relative

**Table 1.** Cellular robustness and immortality correlate with low intrinsic proteome carbonylation levels. For comparison, 10-fold lower protein carbonylation is detected for *D. radiodurans* relative to *E. coli*, as well as in ES cells compared to HeLa cells. MEFs and the embryonic stem cells were a generous gift from Dr Alfonso Bellacosa (Fox Chase Cancer Center). Protein extracts were prepared in 10 mM PBS, pH 7.4, using Dounce homogenizer in the presence of protease inhibitors (Roche). In the case of *E. coli* 1 mg ml$^{-1}$ and for *D. radiodurans* 10 mg ml$^{-1}$ of lysozyme was used. Protein concentration was determined using the Bradford assay and the total protein carbonylation was measured as previously described [9]. Protein carbonylation measurements were performed by previous elimination of lipids and nucleic acids, presumably accounting for discrepancies with some published data.

| cell line | nmol carbonyl/ mg protein |
| --- | --- |
| HeLa cells | 4.97 |
| CHO fibroblasts | 3.88 |
| mouse skin fibroblasts | 9.26 |
| P83 mouse embryonic fibroblasts (MEF), second passage | 4.58 |
| embryonic stem (ES) cells I | 0.51 |
| induced pluripotent stem cells, 10th passage | 2.43 |
| *Escherichia coli* (exponentially growing) | 2.05 |
| *Deinococcus radiodurans* (exponentially growing) | 0.23 |

to *E. coli* and *Deinococcus radiodurans* bacteria. This comparison makes sense since saturation levels of proteome carbonylation are very similar in all tested species (bacteria, yeast, nematodes, rotifers, murine and human cells in culture [9,36]; see also figure 4), revealing similar intrinsic 'carbonylability' of proteomes across the species. However, such carbonylability is readily affected by changing the quality of the proteome (figures 2 and 8).

The lowest protein carbonylation value was observed for *D. radiodurans*, an extremely radiation-resistant bacterium known for its high constitutive antioxidant protection of proteins [30,47], followed closely by mouse ES cells (table 1). A high level of protein carbonylation in mouse skin fibroblasts was reduced by their transformation into iPSC. Human HeLa tumour cell line maintains rather high but constant level of protein carbonylation. Thus, cell immortality (i.e. unlimited division potential) correlates with constant protein carbonylation levels, either low (ES, iPSC) or relatively high (tumours).

Constant levels of protein carbonylation were found along the lifespan of the long-lived (3 decades) cancer- and ARD-resistant naked mole rat [47], which shows constant death rate throughout its lifespan [48]. Unexpectedly, a relatively high, but constant across the lifespan, level of protein carbonylation in naked mole rat is due to high carbonylation of two abundant proteins [49]. Low carbonylation of all other proteins in the naked mole rat can be related to about 10-fold lower error rates in protein biosynthesis [50] and increased intrinsic protein stability when compared with mouse proteins [51]. Reduced intrinsic protein oxidability (carbonylation) is also related to increased stability of proteins in pathogenic bacteria [52].

Tumour, ES and iPSC cells are fuelled mainly by low ROS producing glycolysis. Therefore, it was unexpected that all of about 60 human tumours tested in our laboratory (hepatocarcinoma, pancreatic adenocarcinoma and colon cancer) displayed increased total protein carbonylation, compared to nearby healthy tissue of the same organ (unpublished experiments and figure 6). This hints to low proteome quality in cancers, presumably due to TOR activation (see above), which is in agreement with constitutively increased chaperone levels in, and phenotypic variability of, tumour cells [53]. Such proteomic fragility could inspire therapeutic innovations.

## 2.11. α-synuclein mutations predisposing to Parkinson's disease sensitize α-synuclein to *in vitro* oxidation

We can consider ARD as particular details of the ageing process with snowballing phenotypic consequences first in the affected organ. The intrinsic oxidation resistance of natively folded proteins is fragile (see above). Quantitatively, the impact of intrinsic protein instability, random biosynthetic errors and folding inaccuracy on protein carbonylation is greater than the impact of the variation in ROS [9,10,13,52]. Random translation errors are not frequent enough to affect all molecules of a given protein species, but will produce a variety of amino acid substitutions including those that bestow increased susceptibility to oxidation of individual molecules. However, one and the same consistent protein error—originating from a silent gene mutation—is present in all molecules of the affected protein, and could predispose such proteoform to oxidation.

To test this hypothesis, we studied human α-synuclein, a protein whose aggregation is associated with Parkinson's disease. As disease progresses, α-synuclein aggregates into neuro-toxic Lewy bodies [54]. Analyses of the Lewy bodies from motor neurons of patients with early-onset Parkinson's disease revealed that their α-synuclein carries one of two mutations: A30P (disease onset at age 50–60) or A53T (disease onset around age 30) [55].

We have irradiated *in vitro* the two mutant proteoforms of purified α-synuclein with a range of γ radiation doses and compared their carbonylation with that of the 'wild-type' α-synuclein. We found that both α-synuclein mutants are more sensitive to protein carbonylation (displaying both higher rates and saturation levels) compared with the wild-type, A53T mutant being far more sensitive than the A30P (figure 7). Supposing that error rates in biosynthesis and folding of three commercially available α-synuclein proteoforms are similar, the observed difference in the detected carbonylation between the wild-type and two mutant α-synucleins reflects the difference in their intrinsic susceptibility to oxidative damage. Similar results were obtained by hydrogen peroxide treatment (not shown).

Thus, the intrinsic, mutation-mediated predisposition of α-synuclein proteoforms to *in vitro* carbonylation correlates with predisposition to Parkinson's disease (and the time of its onset), and raises the questions of whether (i) protein sequence polymorphism determines the polymorphism of protein sensitivity to oxidation, and (ii) intrinsic oxidability of disease-relevant proteins causes predisposition to particular ARD. Unless this result with three natural human α-synuclein proteoforms is a unique exception, we have a

royalsocietypublishing.org/journal/rsob Open Biol. **9**: 180249

royalsocietypublishing.org/journal/rsob    Open Biol. 9: 180249

paradigm for the mechanism of predisposition to ARD and for monitoring their progression or regression with the relevant molecular biomarker (that, for Parkinson's disease, can be found in erythrocytes where α-synuclein is abundant).

Now, we can advance the hypothesis that natural protein sequence polymorphism translates into polymorphism of proteomic oxidation patterns of individual proteins reflecting person-specific ensembles of conditional (usually called silent) mutations. Age-related proteome oxidation [8] is expected to keep augmenting the biological (phenotypic) effect of such silent mutations (typically amino acid substitutions) over time and account for inter-individual variation in the onset of ARD and conditions. Ideally, the identification of such polymorphisms should provide for diagnostics, at any age, of individual's predisposition to the first in line (but also second and third in line) disease emerging with age.

## 2.12. Genetics and proteomics of diseases

The proposed concept for the predisposition to ARD integrates genetics and proteomics of human non-infectious diseases (but it includes predispositions to infectious diseases). The disease phenotype of a gene mutation that destroys the function of the encoded protein is considered a syndrome—an inborn disease manifested from the beginning of baby's life. Syndromes are relatively rare due to counter-selection of reduced reproductive fitness, whereas the disease-associated silent polymorphic mutations (like those of α-synuclein in figure 7) are regular predispositions to diseases that emerge typically after the reproductive period (those appearing earlier were subjected to counter-selection). Figure 7 suggests that such mutations are conditional, usually missense, mutations with age (oxidation)-dependent phenotypic expression. Polymorphisms that are phenotypically silent during reproductive period are free to accumulate in the population.

Hence, it appears that the key genetic difference between rare syndromes and the omnipresent predispositions to ARD resides in the nature of relevant mutations that determines their phenotypic expression period. When ageing and ARD appear dissociated, as in old-looking healthy centenarians, it is likely to be due to the chance avoidance of major risk-carrying 'weak links', i.e. oxidation-sensitive proteoforms (polymorphisms), or to an increased defence against oxidation of all proteins.

As shown in bacteria [9], mutations affecting proteins that control the precision of biosynthesis and folding of many (via altered chaperone activity) or all (via altered ribosomal fidelity) proteins should predictably have a systemic effect on oxidative proteome damage (figure 1). The expected consequence of such mutations in complex organisms is the acceleration of ageing and of predisposed ageing-associated pathologies. Only limited phenotypes of this kind will be compatible with survival. Consistent with kinetic proofreading, mutations improving the fidelity of biosynthesis and folding usually display reduced rates of biosynthesis and growth (as in TOR-Off regimen, see above). Since, in growing cells, protein biosynthesis and folding consume presumably over 80% of cellular energy, there is a trade-off between efficacy and robustness. There is selection for efficacy (survival and reproduction), not for perfection.

# 3. Principle of a new preventative and curative medicine

Here, we propose that the root cause of many, perhaps all, ARD is simple, measurable, preventable and sometimes reversible by increased protection [33] and turnover of proteins. To compensate for the malfunction of damaged or lost organs, medicine uses accessory or replacement strategies: mechanical, optical, electronic, chemical (drugs) and biological (e.g. organ, tissue and, recently, cellular transplantation, i.e. cell therapy) prostheses. Organs malfunction when their cells malfunction, and cells malfunction when their proteins malfunction, which breaks ground for a new approach—protein therapy or 'proteomedicine'. Until the advent of a safe massive manipulation of somatic gene sequences, proteins are more credible targets for medical interventions. While artificial protein transplantation is unthinkable, tissue-based cellular parabiosis is a natural way of receiving quality proteins (or their mRNA), or metabolic products thereof, from healthy cells (see the companion paper [29]). Stimulating the recovery of cells' inbuilt capacity to renew their natural proteome homeostasis (de facto cell rejuvenation; see induced pluripotent stem cells in table 1) is both easier and more realistic than hazardous targeted pharmacological interference by 'smart drugs' with highly evolved complex, insufficiently understood, cellular and organismal homeostasis.

Just as 'bad news' increases exponentially with time through a vicious circle, so could 'good news' (reduction in proteome damage) keeps increasing via a virtuous circle. One way is by reducing protein damage and/or stimulating protein turnover (improving the fidelity of protein synthesis and the efficacy of protein turnover by proteasome and autophagy) (figure 1), like after cell reprogramming (table 1) and perhaps during heterochronic parabiosis [56]. Heterochronic parabiosis experiments (connecting the bloodstreams of isogenic old and young mice) showed rapid reversibility of common ageing phenotypes and even the reversion of age-related cardiac and bone pathologies [57] and cognitive impairments [58]. However, the short lifetime of the rejuvenating effects upon interrupted heterochronic parabiosis suggests that defects were compensated, not corrected. There is apparently a 'memory' of the biological age, presumably mainly at the level of epigenetic genome alterations. Reversible phenotypic compensation, downstream of age-related genome alterations, by the mTOR downregulation during heterochronic parabiosis is worth testing.

For all practical purposes, effective 'generic' antioxidants, neutralizers and detoxifiers of ROS should prevent ageing and ARD, especially if taken from early life. Later-life consumption of antioxidants should also be useful, with the exception of already developing cancers (e.g. due to growth stimulation of early-stage tumours by reducing apoptosis and increasing the fitness of tumour cells [59]). Remarkably, two weeks of administration of a spin-trap antioxidant to aged gerbils reduced protein carbonylation in the brain and reversed short-term memory loss, but the effect lasts only during the treatment [33]. The question of whether a lifelong treatment would delay the onset of ageing remains open.

An alternative approach consists of targeting particular proteins for disease prevention and healing. In this paper, we propose that the predisposition to ARD stems (at least sometimes) from the polymorphism of intrinsic protein sensitivity to oxidative damage caused by protein sequence polymorphism (figure 8).

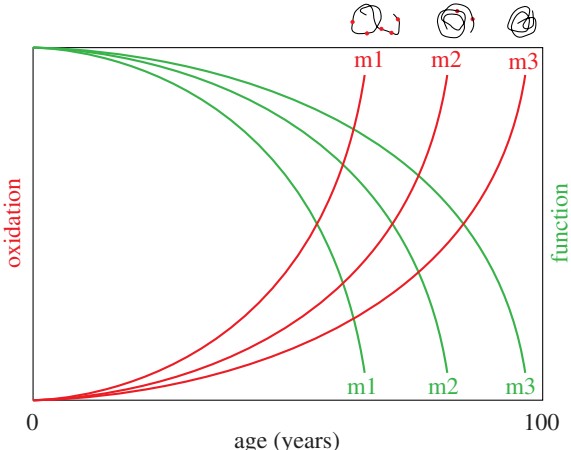

**Figure 8.** Susceptibility of proteoforms to oxidative modifications determines the onset of their oxidation over time and leads to their functional decline. Red dots represent oxidation events. Variants of the same protein (proteoforms) characterized by stable tertiary structure are intrinsically more resistant to oxidative damage and will experience oxidation with a longer delay, compared with disordered/more flexible proteoforms. m1, m2 and m3 stand for three proteoforms (as an interpretation of data in figure 6) and their intrinsic oxidability is symbolized by the drawings above the figure.

Thus, protection and/or restoration of the protein function that is the 'weakest link' in an individual person's health can be a key to prevention or delay of the disease onset. This opens new avenues for the screening, or synthesis, of a new class of molecules with the task to specifically prevent damage to fragile variants of disease-relevant proteins (either as protectors of the site exposed to oxidation or as chemical chaperones—correctors of structure from oxidation sensitive to resistant; figure 3). This is conceptually similar to Vertex Pharmaceuticals' approach to cystic fibrosis (F508del-CFTR structure corrector molecules) [60].

Perhaps only a few hundred specific molecules could both prevent and revert nearly all ARD. Imaginably, sometime in the future, the entire human population could be taking protein polymorphism-specific molecules to suppress the innate fragility (oxidation-sensitivity) of relevant proteoforms. Proteins, rather than genes, could become subject to disease mitigation by protection, correction or reversion of biological function, normally eroding by age-related oxidative damage.

This far-reaching approach aims at a profound change in public health and longevity, and is technically not more demanding than the development of current 'smart drugs'. The possibility that the inborn health-related genetic inequalities could be remedied by protein structure-correcting/protecting molecules (preventing and curing ARD as long as they are consumed) is thrilling. Such direct action upon the cause of degenerative diseases is reminiscent of low-tech prevention and cure of infectious diseases by vaccination and antibiotics, successfully applied before any significant knowledge about infection and immunity. Health-wise, acting 'upstream' upon the cause(s) of disease(s) eliminates the need to study their complicated downstream consequences.

## 4. Closing remarks

Countless publications link the onset of many ARD either with metabolism and ROS activity (oxidative stress) or with protein misfolding and aggregation. We merge here these two phenomenologies in one simple model (figure 3) that integrates TOR signalling (see above) in a way that is compatible with relevant literature. For instance, the severity and mode of misfolding caused by numerous mutations in human SOD1 gene was shown to correlate with destabilization of SOD1 protein and the onset and severity of neurodegenerative ALS disease [61]. Furthermore, specific chaperones (HSP90) can buffer phenotypic expression of Fanconi anaemia disease-causing missense mutations in FANCA protein [62]. Since chaperone activities are in competitive antagonism with protein oxidation upon misfolded proteins (figure 3), pharmacological targeting of misfolded proteins can become the basis of a new medicine. A repertoire of new small molecules acting similarly to natural chaperones [62], and local protectors from ROS activity, could preclude the expression of latent protein defects that cause ARD. By mitigating ARD as the main cause of human morbidity and mortality, new drugs would extend human health span and prolong productive life. Assessing the reality and applicability of this concept requires validation on many ARD. Only a coordinated international effort could provide adequate efficacy.

Identifying underlying cause(s) and mechanism(s) that trigger the onset of ARD is the most important global health project because up to 90% of morbidity and mortality in developed countries, and progressively so in developing countries, is associated with age. The incidence of all ARD (principally rheumatoid, cardiovascular, malignant, neurodegenerative and immunity-related) increases in human population with about fifth power of age. We propose that protein damage is their common root cause and protein sequence polymorphism the likely determinant of the predisposition to ARD via silent mutations that sensitize relevant proteins to oxidative damage.

## 5. Methods related to figure 5

### 5.1. Strains and growth conditions

WT S288C with Hsp104-GFP fusion was obtained from INSERM U1001. Petite strains (mutants devoid of mitochondria) were made as previously described [63]. Briefly, ethidium bromide (10 μg ml$^{-1}$) was added to an exponential culture and left to grow for at least 12 h and plated. After 2–3 days, petite colonies were picked and cultured. Both strains were grown on YPD medium with 2% (w/v) glucose at 30°C with shaking. All experiments were performed on cells from mid-exponential phase: cells were grown until OD 0.6–0.8, harvested by 5 min centrifugation at 4000$g$, washed and treated accordingly.

### 5.2. Gompertz statistical analysis

Regression analysis was performed using R studio software (v. 3.0.2), with the R Gompertz fitting script that contains *grofit* package, which can be found at http://cran.r-project.org/package=grofit. *grofit* package was developed to fit many growth curves obtained under different conditions in order to derive a conclusive dose–response curve. It fits data to different parametric models and in addition provides a model-free spline method to circumvent systematic errors that might occur within application of parametric methods. This attribute increases the reliability of the characteristic parameters (e.g. lag phase, maximal growth rate, stationary phase) derived from a single growth

royalsocietypublishing.org/journal/rsob Open Biol. 9: 180249

curve [64]. For the statistical analysis in the R script, time = c variable was obtained experimentally by micromanipulations as the time of death, while mort = c variable, obtained the same way, represented the percentage of death among yeast cells. Plot (*MyModel*) of the R script outputs three Gompertz parameters (lag phase, maximum growth rate, maximum slope), and their standard error.

## 5.3. Microscopy: slide preparation

Microscope slides were prepared as follows: 150 μl of YPD media containing 2% agarose was placed on a preheated microscope slide, and cooled, before applying yeast cells to obtain a monolayer. The cells were previously centrifuged at 4000*g* for 3 min, and resuspended in 50 μl YPD. Once dry a coverslip was placed and sealed.

## 5.4. Live cell imaging and image analysis for counting protein aggregates

The slide was mounted on Volocity software (v. 6.3; Perkin Elmer)-driven, temperature-controlled Nikon Ti-E Eclipse inverted/UltraVIEW VoX (Perkin Elmer) spinning disc confocal set-up. We also employed the auto-focus system (Perfect Focus, Nikon), and Nano Focusing Piezo Stage (NanoScanZ, Prior Scientific). Images were recorded through 60xCFI PlanApo VC oil objective (NA 1.4) using coherent solid state 488 nm/50 mW diode laser with DPSS module, and 1000 × 1000 pixels 14 bit Hamamatsu (C9100–50) electron-multiplied, charge-coupled device (EMCCD). The exposure time was 300 ms, and 5–10% laser intensity was used. The images were analysed by using ImageJ software. The number of cells with Hsp104-GFP foci was counted manually. A total of 466 WT OGT cells during 12 h and 207 petite OGT cells during 35 h were examined.

Data accessibility. This article has no additional data.

Competing interests. We declare we have no competing interests.

Funding. Swiss Fondation Nelia and Amadeo Barletta funded our research from 2012 to 2015. Ira Milosevic (European Neuroscience Institute, Goettingen) funded (Emmy Noether Young Investigator Award from the Deutsche Forschungsgemeinschaft) and supervised the spinning disc fluorescent microscopy of budding yeast aggregates.

Acknowledgements. We are grateful to Monsieur Jean-Noël Thorel, the Fondation Jean-Noël Thorel, Naos group and Orbico company (Mr Branko Roglic) for sponsoring MedILS and our research on biological robustness, ageing and diseases in MedILS. We thank Thomas Nystrom for the gift of *Escherichia* coli ribosomal fidelity mutants. M.R. thanks Matthew Meselson, David Grainger, Philippe Even, Zoran Dermanovic, François-Xavier Pellay and Adam de Graff for extensive discussions in relation to this paper. Matea Peric, Marina Musa, Anita Lovric and Marina Rudan participated in the measurement of replicative lifespan for budding yeast strains. Matea Peric performed the Gompertz equation fitting. Mikula Radman helped with drawings (figures 1, 3, 4 and 8).

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
