## [Reviewer comments · Open Biology]

Review History

RSOB-18-0249.R0 (Original submission)

Review form: Reviewer 1

Recommendation

Accept with minor revision (please list in comments)

Are each of the following suitable for general readers?

- a) **Title**
Yes
- b) **Summary**
Yes
- c) **Introduction**
Yes

Is the length of the paper justified?

Yes

Should the paper be seen by a specialist statistical reviewer?

No

Is it clear how to make all supporting data available?

Not Applicable

Is the supplementary material necessary; and if so is it adequate and clear?

Not Applicable

Do you have any ethical concerns with this paper?

No

Comments to the Author

This is an interesting 'hybrid' paper, which mostly read as a review with some new datasets added to it to show or support a concept. The experiments and datasets provided are fitting nicely with the theories presented, which are focused on the hypothesis that aging, and age-related disorders, can be explained in terms of protein misfolding (by damage/ mistranslation) and silent mutations (polymorphisms) sensitizing proteins to oxidation (for example by carbonylation) creating a 'snowballing' phenotype. The paper is conceptually intriguing and the authors have nicely summarized a broad field of research conducted during many years into a coherent theory of aging.

I have but a few comments and questions the authors might want to consider:

1. The authors argue that chaperones might be exquisitely sensitive to targets for carbonylation. This notion is actually supported by data also from embryonic stem cells (Hernebring, et al., 2006 Proc. Natl. Acad. Sci. USA), drosophila (Fredriksson et al., 2012 Aging Cell), and plants (Johansson et al., 2004 J. Biol. Chem) and Meisner et al., (2005, Nature Genet) showed that overproduction of a chaperone mitigates the deleterious effects of increasing mutations, which seems to be in line with the theories presented in this paper.
2. The suggested link between ROS and TOR is interesting and I wonder if the authors could do a similar comparison and link in bacteria, between in this case ROS/damage and the stringent response?
3. On page 14 it is stated that the accumulation of Hsp104-containing aggregates is one of the best described aging phenotypes. The papers cited (36, 37) might not be the most proper here as Hsp104 accumulation at sites of carbonylated proteins in aged cells were first described by Erjavec et al., 2007 Genes Dev. Paper 36 aimed at demonstrating that aggregates diffuse freely but slowly which would explain asymmetrical inheritance of aggregates during cytokinesis, a notion that was later refuted.
4. Figure 5. In this figure it is shown that rho0 mutant display an extended lifespan. There have been reports on the opposite and also reports on no effect of a rho0 mutation. The problem is that rho0 cells have been generated by a number of different ways and targeting different genes. Perhaps a short comment on how their rho0 mutant was generated could help the reader here.
5. One major theme of this review/paper is that oxidation of an already misfolded protein creates a snowballing effect that might drive the aging process. A key question that emerges is; is oxidation really required for the phenotypic penetrance discussed or could misfolding be enough (through aggregation for example). Perhaps the authors could mention experiments supporting that oxidation is required or if such data is lacking how one could approach this chicken and egg problem.

Decision letter (RSOB-18-0249.R0)

28-Jan-2019

Dear Professor Radman,

We are pleased to inform you that your manuscript RSOB-18-0249 entitled "Protein damage, ageing and age-related diseases" has been accepted by the Editor for publication in Open Biology. The reviewer has recommended publication, but also suggest some minor revisions to your manuscript. Therefore, we invite you to respond to the reviewer's comments and revise your manuscript.

Please submit the revised version of your manuscript within 14 days. If you do not think you will be able to meet this date please let us know immediately and we can extend this deadline for you.

- 1) A text file of the manuscript (doc, txt, rtf or tex), including the references, tables (including captions) and figure captions. Please remove any tracked changes from the text before submission. PDF files are not an accepted format for the "Main Document".
- 2) A separate electronic file of each figure (tiff, EPS or print-quality PDF preferred). The format should be produced directly from original creation package, or original software format. Please note that PowerPoint files are not accepted.
- 3) Electronic supplementary material: this should be contained in a separate file from the main text and meet our ESM criteria (see <http://royalsocietypublishing.org/instructions-authors#question5>). All supplementary materials accompanying an accepted article will be treated as in their final form. They will be published alongside the paper on the journal website and posted on the online figshare repository. Files on figshare will be made available approximately one week before the accompanying article so that the supplementary material can be attributed a unique DOI.

Online supplementary material will also carry the title and description provided during submission, so please ensure these are accurate and informative. Note that the Royal Society will not edit or typeset supplementary material and it will be hosted as provided. Please ensure that

the supplementary material includes the paper details (authors, title, journal name, article DOI). Your article DOI will be 10.1098/rsob.2016[*last 4 digits of e.g. 10.1098/rsob.20160049*].

4) A media summary: a short non-technical summary (up to 100 words) of the key findings/importance of your manuscript. Please try to write in simple English, avoid jargon, explain the importance of the topic, outline the main implications and describe why this topic is newsworthy.

Images

Data-Sharing

It is a condition of publication that data supporting your paper are made available. Data should be made available either in the electronic supplementary material or through an appropriate repository. Details of how to access data should be included in your paper. Please see <http://royalsocietypublishing.org/site/authors/policy.xhtml#question6> for more details.

Data accessibility section

Sincerely,

The Open Biology Team
<mailto:openbiology@royalsociety.org>

Reviewer's Comments to Author:

This is an interesting 'hybrid' paper, which mostly read as a review with some new datasets added to it to show or support a concept. The experiments and datasets provided are fitting nicely with the theories presented, which are focused on the hypothesis that aging, and age-related disorders, can be explained in terms of protein misfolding (by damage/mistranslation) and silent mutations (polymorphisms) sensitizing proteins to oxidation (for example by carbonylation) creating a 'snowballing' phenotype. The paper is conceptually intriguing and the authors have nicely summarized a broad field of research conducted during many years into a coherent theory of aging.

I have but a few comments and questions the authors might want to consider:

1. The authors argue that chaperones might be exquisitely sensitive to targets for carbonylation. This notion is actually supported by data also from embryonic stem cells (Hernebring, et al., 2006 Proc. Natl. Acad. Sci. USA), drosophila (Fredriksson et al., 2012 Aging Cell), and plants (Johansson et al., 2004 J. Biol. Chem) and Meisner et al., (2005, Nature Genet) showed that

overproduction of a chaperone mitigates the deleterious effects of increasing mutations, which seems to be in line with the theories presented in this paper.

2. The suggested link between ROS and TOR is interesting and I wonder if the authors could do a similar comparison and link in bacteria, between in this case ROS/damage and the stringent response?

3. On page 14 it is stated that the accumulation of Hsp104-containing aggregates is one of the best described aging phenotypes. The papers cited (36, 37) might not be the most proper here as Hsp104 accumulation at sites of carbonylated proteins in aged cells were first described by Erjavec et al., 2007 Genes Dev. Paper 36 aimed at demonstrating that aggregates diffuse freely but slowly which would explain asymmetrical inheritance of aggregates during cytokinesis, a notion that was later refuted.

4. Figure 5. In this figure it is shown that rho0 mutant display an extended lifespan. There have been reports on the opposite and also reports on no effect of a rho0 mutation. The problem is that rho0 cells have been generated by a number of different ways and targeting different genes. Perhaps a short comment on how their rho0 mutant was generated could help the reader here.

5. One major theme of this review/paper is that oxidation of an already misfolded protein creates a snowballing effect that might drive the aging process. A key question that emerges is; is oxidation really required for the phenotypic penetrance discussed or could misfolding be enough (through aggregation for example). Perhaps the authors could mention experiments supporting that oxidation is required or if such data is lacking how one could approach this chicken and egg problem.

Decision letter (RSOB-18-0249.R1)

28-Feb-2019

Dear Professor Radman,

We are pleased to inform you that your manuscript entitled "Protein damage, ageing and age-related diseases" has been accepted by the Editor for publication in Open Biology.

Sincerely,

The Open Biology Team
mailto: openbiology@royalsociety.org